# Autism Spectrum Disorder (ASD) and the Family Inclusive Airport Design Experience

**DOI:** 10.3390/ijerph18137206

**Published:** 2021-07-05

**Authors:** Monica Cerdan Chiscano

**Affiliations:** Economics and Business Studies, Open University of Catalonia, 08018 Barcelona, Spain; mcerdanc@uoc.edu

**Keywords:** inclusive research, public health, airport management, service experience innovation, co-creation, families of children with autism spectrum disorder

## Abstract

The literature on air travellers with psychiatric disorders is limited. This perspective article highlights various travel-related aspects of autism spectrum disorder (ASD). The airport experience can be stressful for families of children with autism spectrum disorder (FwASDs). The aim of this study was to explore the airport experience of FwASDs using the value co-creation process approach to assist airport managers in designing improved experiences for this specific passenger segment. This study responds to the current climate in which airports are developing awareness programmes in relation to children who require special assistance at airports. The prevalence of children with ASD is 1/68. While a number of airports throughout the world have adopted procedures addressing the needs of those with cognitive impairment, these advances are far from universal. As part of an academic–industry collaboration between Vueling airlines and the Spanish airport operator Aena, 25 FwASDs took part in an inclusive airport research project in the city of Barcelona from November 2015 to April 2016. Employing a qualitative methodology that incorporated focus groups, ethnographic techniques, and post-experience surveys, the study contributes to extending the body of knowledge on the management of the value co-creation process for challenging passenger segments within the airport context. The study explains how ensuring adequate resource allocation to this passenger segment can improve the family-inclusive design of the airport experience and offers managerial recommendations.

## 1. Introduction

Travelling can be challenging for children and adults with autism. The changes in routine, unpredictability, crowds, and new noises and sights can all make the experience difficult for children with ASD and their families. The literature on international travellers with psychiatric disorders is limited [1,2]. Public health officials should be aware of the unique needs of travellers with ASD when providing pre-travel health advice [2]. People living with mental illness are among the most vulnerable groups of international travellers [3,4]. The growing market segment of passengers with autism, a neurodevelopment condition that affects a person’s ability to properly communicate and relate to their environment and other people [5], presents a particular challenge for airports, as passengers with ASD exhibit particular difficulties during the air travel experience and require specific kinds of support to be able to travel at ease. Air travel can be exhausting, from check-in to onboard service, for this passenger segment and their families [6]. According to [7], passengers with mobility and non-mobility disabilities requiring service assistance support at airports comprise a major passenger segment worldwide.

The literature to date has mainly focused on inclusion and accessibility for passengers with mobility disabilities in the air travel experience. Recently, however, attention has shifted towards the passenger segment with non-mobility disabilities, known as hidden disabilities, such as cognitive impairments including autism [8].

In 2014, the World Health Organization [9] called for the access needs of people with ASD to be properly addressed. The travel industry and public health sector, therefore, should endeavour to eliminate the major constraints encountered by passengers with autism [2,3,4,5,6,7,8,9,10], yet research on international travellers with developmental or cognitive disabilities is limited [2]. Compared with the general population, children diagnosed with autism spectrum disorder (ASD) tend to have deficits in adaptive skills, which makes novel environments such as an airport very difficult to manage [11,12], and the airport experience for a child with ASD and their family can therefore be overwhelming [10]. Without appropriate preparation, going through an airport and boarding an aircraft can be unpredictable and anxiety-provoking for both the child with ASD and their family [13]. There is therefore an onus on public health officials, airport operators, and airline companies to respond quickly with a design of airport experiences for all.

Travel is a complex practice that requires an understanding of the related embodied experience within the travel planning process [14] and continues to pose numerous challenges for passengers with mobility or non-mobility disabilities [15]. As a result, awareness programmes, such as Open Days, have emerged in the airport context to provide parents with the opportunity to see whether their child is able to successfully negotiate the demands of the air travel process and in what specific areas their child may require additional time, support, and preparation [6,7,8,9,10,11,12,13,14,15,16]. At the same time, these programmes are intended to improve airport staff’s knowledge of travellers with ASD.

While these inclusion resources are extremely important for improving the airport experience of FwASDs, there is a need to design a methodology of inclusion and accessibility at airports that enables these programmes to become a means of generating ideas. One way of addressing the issue is to improve the understanding of the needs of FwASDs by actively engaging them in the service delivery process [17] according to the service-dominant logic approach to value co-creation [18,19]. In service-dominant logic, value [18] is created collaboratively by and for both the customer and the company by means of peer contributions from different stakeholders [20]. We used the definition of [21], which defines stakeholders as individuals or associations capable of influencing the company’s results.

The service-dominant logic value co-creation process has rarely been applied in the literature on air travel management [22] with respect to challenging markets. The aim of this study was to address this deficit to help improve the airport experience for FwASDs. We drew on the explanation of [17] for how to manage the value generation process in co-creation within the framework of service-dominant logic [18]. Our focus was on touchpoints or encounters—the processes of interaction that occur between the passenger and the company that when properly managed provide opportunities for value co-creation. Further empirical research is needed to understand how to manage the value co-creation process [23] when the passenger requires special airport service support. The lack of previous empirical studies is partially explained by the fact that the benefits of the process are not easy to measure [24].

This paper reports on the “*Alas para el autismo*” (https://www.youtube.com/watch?v=ykQPdmO5v2g (accessed on 27 April 2016 )) project undertaken at the Barcelona-El Prat airport in the city of Barcelona between November 2015 and April 2016. The project aimed to contribute to the body of knowledge of the value co-creation processes and the management of air travel involving FwASDs. Its objective was to improve the airport experience for FwASDs while providing airport operators and airline company managers with insights that enable them to identify the proper resources that, when faced with critical encounters with FwASDs, could lead to more accessible and inclusive value experiences for FwASDs. In addition, this study explains the benefits of the process for the airport operators, airline companies, and passengers with autism when the encounters are appropriately designed, and the learning and communication needs presented by FwASDs are taken into consideration. To do so, we aimed to explore what [25] introduced as “critical encounters”. As a result, this study will orient airport managers towards the use of appropriate resources and ensure that new resources are used effectively [26] to promote inclusion.

### 1.1. Barriers to the Airport Experience for FwASDs

Interestingly, [27] found that travel enhances children’s personal development, boosts their knowledge and capacity for understanding, and provides communicational and relational skills. Moreover, the perceived benefits extend to the family sphere, reducing stress and anxiety and improving the family’s quality of life [28,29]. Nevertheless, air travel is a challenging activity for many FwASDs. The high vulnerability of children with disabilities means that their families’ travel intentions are influenced by the perception of certain travel-related constraints, such as fear of air travel due to the condition of the children with ASD, which impedes or prevents children with disabilities from taking part in travel activities [30]. However, according to [31], for the family of a child with disabilities, expectations about the travel experience are the key condition for the intention to travel. Travel is especially challenging for FwASDs when the means of transportation is air travel, and the possible presence of elements that limit or prevent the participation of the child will generate perceived helplessness on the part of the family and affect their intention to travel [32]. Many parents will not travel by aeroplane with their children out of the fear that their child will not be able to negotiate the airport without an extremely high level of distress.

In addition to constraints that prevent the participation of children with disabilities, a family’s intention to travel can be influenced by other personal fears, especially those related to personal and health security [33]. This fear is currently one of the most worrisome, not only for FwASDs but for the entire population.

Moreover, while an increasing number of airports are adapting new procedures to address the accessibility issues of passengers with mobility and non-mobility disabilities, including FwASDs, these have not proven to be universal advances [8].

Despite advances in universal design—the systemic process of adapting and creating new products and services for all [34]—and accessibility standard regulations, barriers to accessibility remain in aircraft and airports. Numerous air travel-related challenges for passengers with disabilities have recently been identified as deserving of research attention [8,14,35,36]. These include communication barriers (inappropriate visual cues, non-adapted signage, and lack of information provision in adapted formats, making the information inaccessible), service barriers involving transport staff (inappropriate staff attitudes), and environmental barriers (primarily excessively noisy concourses and ineffective wayfinding systems). Unfamiliar experiences and uncontrollable factors, such as delays and cancellations, pose a particular challenge for passengers with ASD and their families and may result in additional anxiety and stress during air travel. It has also been reported that, when travelling, FwASDs confront challenges, such as ensuring that the appropriate services are available, behavioural problems associated with queuing for long periods of time, and social exclusion [27,37,38,39]. Given that many FwASDs are still affected by these barriers, we used the value co-creation framework [18] to explore critical encounters with the objective of designing more inclusive airport experiences for FwASDs.

Interestingly, the allocation of proper resources to critical encounters has been demonstrated in the literature to be extremely important for ensuring a successful user experience. The literature on value experience has shifted its focus on resource integration [17], specifically the study of how certain resources provided by companies can lead users to generate positive outcomes [40,41]. For instance, a children’s storybook explaining the airport experience as a potential resource allows the child to process one concept at a time and understand the appropriate behavioural response required in the situation at his or her own pace [42]. Many parents and professionals suggest that a family preparing to travel by aeroplane contact the airport operators and airline companies in advance to inform them about their ASD child’s communication abilities and possible challenges that may arise [43]. Another relevant practice for enhancing the passenger experience is to train airport and aircraft staff in the basics of ASD and strategies to facilitate sensitive and successful interactions throughout the airport experience [44].

### 1.2. Service-Dominant Logic, Value Co-Creation, and FwASD Barriers in the Airport Context

We drew on the [45] study, which found that for successful improvements to occur, the transportation system should engage in continuous collaboration with passengers in the design of experiences. Thus, to acquire knowledge about their passengers, airport operators and airline companies should actively interact with them instead of designing air travel experiences based on their own perceptions. Each passenger possesses unique knowledge about their own travel needs [46].

One strategy being embraced by airport operators and airline company managers is the use of value co-creation to design positive air travel experiences. Interestingly, the framework of [17] is helpful for understanding, testing, and implementing improvements using service-dominant logic and only a minimum number of resources from the company [47]. The co-creation process entails users’ active participation and collaboration with the service provider, all the way from identifying a challenge to implementing and tracking the performance of its solution. On that basis, by fully collaborating with end users and taking their access needs into account, providers are able to personalise the service they offer without investing many resources.

Interestingly, products and services developed in co-creation with users scored higher in terms of satisfaction [48,49] and at the same time used fewer resources throughout the process. Value co-creation can create positive value outcomes [50], including emotional value [51] and social and functional value [52]. Additionally, [24] demonstrated that designing in collaboration with users generated more value than designing in collaboration with experts. However, if the process is not effectively managed, such as in certain cases in which FwASDs need to make special efforts to be involved in the process [53], for example, finding personal free time to participate in focus groups, the participants’ higher expectations of the outcome can lead to dissatisfaction [54].

We also drew on [17] for the identification of the three main types of encounters involved in the value co-creation process: (1) communication encounters or activities involving interaction and communication with passengers, e.g., an adapted storybook or website information adapted to passengers’ communication needs; (2) usage encounters or practices involving the use of the facilities and services provided by the airport operators and airline companies to passengers, e.g., use of lifts, signage, and wayfinding; and (3) service encounters or interactions between the passenger and airport operators and the airline companies’ staff or other passengers.

Although a majority of travel operators are embracing the value co-creation process as a strategic tool, the “how” and “why” are not yet clear [24]. In this study, therefore, we aimed to shed light on the management of value co-creation between FwASDs, airport operators, and airline companies to create value propositions for accessibility. To do so, we drew on the service-dominant logic experience model [19,55,56] as an effective tool. In this model, FwASDs become an operant resource with the capacity to create value experiences through interaction with airport operators and airline companies [57], and airport operators and airline companies can make more effective investments with a maximum delivery of value to FwASDs. In line with [17], we aimed to develop new ways of understanding how airport operators and airline companies can effectively use the service-dominant logic co-creation process through empirical analysis of critical encounters, which have notably received little research attention to date in the context of transportation [58] and specifically airport experiences; hence, we explored a framework for improving the airport experience for FwASDs. This is a gap in the previous research that the present study sought to address.

While previous studies have explored value co-creation in the context of transportation (notably, [48,49,50,51,52,53,54,55,56,57,58,59]), it is rare to find empirical research on co-creating [60,61] in collaboration with FwASDs in the airport context. In this study, we sought to provide empirical evidence to aid the management of the co-creation process and address FwASDs’ airport experience challenges by redesigning the current accessible service delivery to FwASDs with a simple idea-generating exercise.

We therefore proposed the following two research questions:

RQ 1: Which critical encounters—communication, usage, or service—within the co-creation process generate the most value for FwASDs to ensure successful inclusive airport experience design?

RQ2: Which encounter improvements ensure a positive airport experience for FwASDs?

The paper is structured as follows. The introduction is followed by a presentation of the methodology. The results are then presented, followed by a conclusion and discussion of the practical implications. Finally, the limitations of the study and future lines of research are presented.

## 2. Materials and Methods

Our objective was to determine the critical encounters that could add value [25] in the process of designing inclusive and accessible airport experiences for FwASDs. Drawing on the framework of [17] and using an ethnographic methodology, the study consisted of designing airport operator and airline company experiences with the collaboration of FwASDs.

In order to underpin the FwASD airport experience, we used a qualitative approach with a mix of techniques [18,62,63], such as collaborative focus groups with stakeholders [64,65]. These were the Spanish operations director of Aena, the Vueling operations director for the Barcelona-El Prat Airport in Barcelona, two representatives of an association for people with ASD (Aprenem), and three FwASDs. Ethnographic techniques were used to shadow participants during the airport visit experience, in which 25 FwASDs took part, in addition to post-experience surveys.

### 2.1. Project Phases

The research project took place at Barcelona-El Prat Airport and consisted of three phases:

Phase 1: Pre-airport visit experience (from November 2015 to March 2016). Two focus groups (two hours each) of researchers and stakeholders to generate new ideas to be implemented and tested in Phases 2 and 3.

Phase 2: During the airport experience. A lead researcher and two assistant researchers observed and shadowed the 25 FwASDs during the airport visit experience at Terminal 1 of the airport (4 April 2016).

Phase 3: Post-airport experience survey with a 30 min semi-structured questionnaire completed by 10 of the participating 25 FwASDs at the exit of Terminal 1 of the airport (4 April 2016).

#### 2.1.1. Phase 1: Focus Groups

The agenda of the two focus groups was to generate and discuss new ideas to improve airport accessibility for FwASDs, which would be implemented and tested for suitability during Phases 2 and 3. The focus groups’ objective was to explore and identify innovative solutions, such as new ideas to facilitate interaction with FwASDs in encounters that could be critical, including proper airport information provision, usage to facilitate accessibility, and accessible services and facilities within the airport context. New ideas proposed by stakeholders were considered worthy of further testing in Phases 2 and 3 during the airport visit experience. For instance, in Phase 1, the second focus group proposed a storybook adapted to the communication needs of children with ASD in anticipation of the airport visit experience. These guides were designed and produced by the researchers. The use of stories has been shown to help children with ASD to prepare for a novel event and promote greater participation in the event [66]. When preparing children with ASD for an event with a story about it, children are able to focus more on participating in the event and less on processing the new information. Twenty staff members of the airline company staff and airport operator, including security station staff) were provided with a one-day training programme at the airport the day before the visit to improve their understanding of the characteristics of FwASDs and how to cater to their access needs in the airport context. The agenda of the training programme covered the medical and social model of disability, the explanation of the characteristics of FwASDs, the history of autism, the needs of this group in an airport context, and strategies for airlines and the airport staff to deal with their needs. The training was provided by psychologists who were experts in autism from a university. The same airport operator and airline company staff trained in Phase 1 took part in Phase 2 of the project.

#### 2.1.2. Phase 2: Airport Visit Experience for FwASDs

We drew on previous awareness programmes, such as Open Days experiences, conducted at other international airports (Wings for Autism, The Arc, 2012 [67]) for the design of Phase 2. The airport visit experience took place at Terminal 1 of Barcelona-El Prat Airport on 2 April 2016 and lasted for two hours. It consisted of 25 FwASDs taking part in an authentic airport access experience at Barcelona’s airport. The typical airport experience began with the FwASDs entering the airport, going through the check-in process, going through airport security, and waiting to board. Afterwards, the FwASDs were able to board the plane and take their seats in a real-life air travel experience. The team consisted of an actual Vueling crew and aeroplane, security station staff, 2 Aena directors, 2 Vueling directors, 15 volunteers providing support for the visit itself, and 1 lead researcher and 2 research assistants engaged in the data collection process during the airport visit experience. Ideas for improving accessibility identified in Phase 1 were implemented and tested during the airport visit experience. These included a priority check-in service to avoid FwASDs queuing for long periods of time and a quiet room provided for FwASDs, modifying the airport’s current accessibility. Figure 1 represents the map of the airport visit experience. 

#### 2.1.3. Phase 3. Post-Experience Interviews with Participants Using Semi-Structured Questions

At the end of the airport visit experience, having exited the Vueling aircraft, 12 participants willing to answer questions were surveyed individually.

### 2.2. Participants

Table 1 shows the characteristics of the sample (participants with disabilities).

Informed consent was signed by all participants.

### 2.3. Data Collection

We drew on [17] to delimit the scope of the study according to the concept of critical encounters (communication, usage, and service).

#### 2.3.1. Phase 1: Pre-Airport Visit Experience: Focus Groups and Encounters

Focus groups 1 and 2 were fully recorded and transcribed [68]. The project design was adjusted several times in the co-creation process to ensure innovative outcomes, and by the end of focus group 2, the process of generating new ideas failed to produce further discussion.

#### 2.3.2. Phase 2: During the Airport Visit Experience

During the airport visit experience, the participants’ behaviours, opinions, and emotions were collected by researchers, who shadowed them throughout the airport visit journey [62] using observation techniques [69]. More than 60 photographs and handwritten notes were collected, in addition to video recordings.

#### 2.3.3. Phase 3. Data Obtained from Post-Experience Interviews with Participants Using Semi-Structured Questions

Each post-airport experience survey took 30 min and was recorded in full. The questions were based on previous studies in the literature [8,9,10,11,12,13,14] on accessibility and inclusive resources in the airport context. The questions aimed to explore the participants’ perceptions of the critical encounters or touchpoints throughout the airport visit experience.

After data collection in each phase, the researcher reporter had enough insight to answer the research questions.

### 2.4. Data Analysis

Upon completion of the fieldwork, a qualitative data analysis sotfware ATLAS.ti was used to create a relational map by coding the data [70]. In order to gain validity, selected coding outputs were analysed in accordance with previous studies [71]. A five-step qualitative thematic analysis [72] was used. Thus, three categories of critical encounters emerged from the selective coding process:(a)communication (e.g., communication adapted to children with ASD, information provision, signage system, technological advances in communication);(b)usage (e.g., perceived level of physical accessibility, physical space, level of security, and sensory elements of the environment); and(c)service (e.g., interaction with transport staff, service provision for disability, etc.).

We drew on [17,73] to guide our development of axial coding. Table 2 shows the coding process.

## 3. Results

This study sought to address the many challenges faced by FwASDs in their airport experiences. Specifically, we sought to understand which critical encounters generate the most value for FwASDs to ensure their successful family-inclusive airport experience design. To do so, we drew on the framework of [17], identifying and testing the critical encounters with FwASDs in the airport context in order to frame a process for improving the airport experience design strategy for this under-researched passenger segment.

### 3.1. Responses to Research Questions

#### 3.1.1. Response to RQ 1

RQ 1: Which critical encounters—communication, usage, or service—within the co-creation process generate the most value for FwASDs to ensure a successful inclusive airport experience design?

Our results indicate that the three critical encounters identified (communication, usage, and service) should be adapted and are relevant to the co-creation process. All three were scored as equally important in addressing the challenge of accessibility for FwASDs.

The critical encounters for accessibility for FwASDs in the airport system are detailed below:(a)Communication critical encounters. Communication aids for improvement: (a) provision of information in an adapted format on the characteristics of the visit to the airport and the level of accessibility for mobility and non-mobility disabilities, (b) timely, online information in plain language and a visual format of the phases of the airport visit (storytelling and guides in adapted formats, available inclusive resources information, etc.), and (c) potential use of new technologies.(b)Usage critical encounters. Usage features for accessibility improvements: (a) proper allocation in place of inclusive family resources for special needs passengers, such as sensory rooms and priority check-in and boarding.(c)Service critical encounters: Critical service encounters improvements: (a) Accessibility training for airport operators and airline company staff and (b) guidelines with instructions for staff on how to deal with special needs passengers.

Table 3 shows the main potentially inclusive resources that, when properly allocated, were shown to generate positive value outcomes during the FwASD airport visit experience. These resources were identified in Phase 1 and developed by the researchers and stakeholders before the airport visit took place. These resources were tested in use in Phase 2 during the airport visit by FwASDs, and they were assessed for value outcomes in Phase 3 after the visit.

From the data analysis, we identified the following critical encounters for accessibility for FwASDs in the airport system: (a) communication critical encounters, (b) usage critical encounters, and (c) service critical encounters.
(a)Communication critical encounters

The data analysis provided enriching insights into good practices for improvements in communication encounters between the airport operators and airline companies and the FwASDs.

In order to address the lack of adapted information provision identified in previous studies reviewed by [8], new ideas identified by stakeholders in the focus groups in Phase 1 were tested during the airport visit experience in Phase 2, and this was shown to be extremely important when designing airport encounter experiences for FwASDs. For instance, one of the new ideas to emerge from the focus groups was the design of a storybook for parents that enabled them to anticipate the airport experience and put their children at ease about the visit prior to it taking place (see Figure 2). The storybook was developed by the researchers, produced by the stakeholders, and given to FwASDs as a potential resource. The visual features, plain language, and use of pictograms helped children with ASD understand the appropriate behavioural response required by the situation at his or her own pace [42].

The storybook given to each family before the experience took place was tested and regarded positively as an important resource for ensuring a positive experience for FwASDs. In sum, the inclusive resources proved to be extremely useful for passengers with autism, illustrating that effective communication gives FwASDs a clearer idea of what to expect during an airport experience. It was also found that timely information is perceived as important and that these inclusive resources need to be available online.
(b)Usage critical encounters

The data analysis confirmed the challenges faced by FwASDs during an airport experience that were identified in previous studies reviewed by [8], such as a lack of easy wayfinding systems for navigating the airport in general and boarding in particular and the need for accessible service support at airports and on aircraft. To address such challenges in usage critical encounters, the following ideas emerged from the focus groups with stakeholders and were tested during the airport visit, creating positive outcomes for FwASDs.

The results showed that the perceptions of FwASDs in the study show that certain sensory components in the physical environment of the airport system are not appropriate for children with ASD (e.g., large noisy spaces, long queues, difficult wayfinding), as the airport was almost wholly adapted for people with physical disabilities. The data analysis identified the following improvements for the usage design encounters. As identified by [2], queuing at airport security stations may be overwhelming for some children with ASD, and security procedures such as the requirement for children to put their assistive technologies (smartphones, tablets, etc.) through the baggage scanner may be poorly tolerated by these children. Furthermore, unfamiliar environmental contexts such as an airport can lead children with ASD to experience anxiety. An easier wayfinding system for navigating the airport was therefore developed and provided to the FwASDs prior to the visit, and this was shown to be very helpful for their visit. The ideas for inclusive resources that emerged from the focus groups in Phase 1, such as the availability of a sensory room or quiet space, new priority access for check-in and security stations for FwASDs, and an easier wayfinding system for navigating the airport, were implemented in Phase 2 and shown to be of great value in helping the children and their families feel calm during their airport experience.
(c)Service critical encounters

Service was found to be a particularly critical encounter, more so for passengers with autism, who were likely to suffer a great deal of stress, especially when travelling for the first time. Some of the FwASDs reported inappropriate attitudes from staff in their previous experiences at the airport. The results showed that staff, particularly the security stations officers in past experiences reported by FwASDs, were not aware of how to properly communicate with FwASDs (inappropriate use of language, lack of awareness of autism, lack of empathy towards FwASDs, etc.):

At the security station you are required to put your things through a baggage scanner and pass through the security control, but separating the children from their assistive iPad is too challenging. Security airport staff do not always seem to have patience with our special situation and this makes us feel very anxious […].(Mother of child with autism).

These findings are in line with the results of [74]’s survey, which showed that in the airport context, the security area is one of the most challenging practices for people with disabilities. In [74]’s survey, 94.7% of respondents reported finding the security area the most challenging area of the airport experience.

Airport and aircraft staff had previously been identified as being in need of appropriate training on how to interact with FwASDs [75]. They also need to be provided with guidelines for communicating with passengers with ASD. A one-day training programme was therefore provided to 20 airport and aircraft staff members taking part in the airport experience prior to the visit. The results proved very positive for the staff as well as the FwASDs:

Once I was approached by a group of people with intellectual disability and I really didn’t know how to deal with the situation, it made me feel really uncomfortable.(Airport security officer).

This one-day training course has really helped me to understand the needs of these families and I think I am prepared to deal with the problems that may arise […] This is my first time receiving autism awareness training and it has been really interesting.(Airport security officer).

The results related to the service encounters demonstrate the importance of interaction with airport and aircraft staff in reducing barriers, and the airport and aircraft staff played a crucial role in generating new ideas in the focus groups:

We think it would be useful for us airport security staff to have guidelines on how to properly communicate with people with autism, since we don’t have this knowledge […].(Airport security officer).

#### 3.1.2. In Response to RQ 2

Which encounter improvements ensure a positive airport experience for FwASDs?

Participants who were involved from the first phase of the project reported that generating new ideas was a far from a daunting process and produced ideas for design encounters with positive outcomes for FwASDs. It was demonstrated that a learning process took place between FwASDs and the rest of the stakeholders, with ideas coming from both sides and, importantly, from FwASDs as operant resources. Giving a voice to FwASD was shown to generate a mutual process of interaction, dialogue, and learning throughout the co-creation process.

We gave our opinion about the service support and related resources that were missing and it was very comforting having specialists involved who were taking our proposals seriously and developing new improvements, such as an adapted quiet room facility for our visit experience […].(Participants with a 10-year-old child with ASD).

Involving and giving a voice to the families also ensures that their expectations of their voluntary participation are met. Accordingly, stakeholders need to ensure participant satisfaction on the part of the families by implementing and testing in Phase 2 a selection of the ideas that emerge from Phase 1 focus groups. There was a significant risk of increased dissatisfaction on the part of the FwASDs given the enormous mobility effort required for them to attend the airport visit experience. In our study, 92% of the participants reported in Phase 3 that they were quite or very satisfied with the airport visit experience.

Our results confirm that the process of co-creation generates value for FwASDs when FwASDs are involved in the design process prior to the visit.

We can therefore conclude the following:-Adapting communication encounters to FwASDs in the experience design makes the FwASD an operant resource.-Designing critical encounters helps mitigate airport constraints for FwASDs.

## 4. Discussion

Public health officials should be aware of the unique needs of travellers with ASD when providing pre-travel health advice [2]. Nevertheless, despite increased research into the barriers encountered by FwASDs in their air travel experience, there is still little empirical research to guide the management of value co-creation in the design of FwASD-inclusive airport experiences.

Thus, while a number of airports worldwide have adopted procedures addressing the needs of people with cognitive impairment disabilities, barriers to inclusion in the airport experience are still very much an issue for FwASDs.

We have seen that, while inclusive resources at airports, such as Open Days and awareness programmes, are extremely important, it is crucial for the inclusion of passenger segments such as FwASDs that airport operators and airline company managers embrace the value co-creation marketing approach and continually introduce adjustments to the design of their airport service support.

In this study, we demonstrated that improvements in the service support delivery for FwASDs are feasible by means of active interaction between the airport operators and airline companies and the FwASDs, and not only through costly investment. Our findings are in line with [35], which demonstrated that focusing on the target can help mitigate barriers to inclusive travel.

Tellingly, the complexity of designing successful accessibility encounters with FwASDs requires genuine collaboration on both sides. We have shown that, beyond simply making service adjustments that take their opinions into account, this process requires authentic understanding and collaboration with FwASDs. This finding is in line with [35], which showed that without an improved understanding of accessibility challenges and practices, this passenger segment will continue to encounter accessibility issues. For instance, in order for FwASDs to become an operant resource throughout the co-creation process [57], they need to be provided with adapted information and specific inclusion resources to avoid the negative outcomes described by [53].

We also explored how effective communication encounters during the design of the experience can lead to an interactive learning process [76,77,78] since airport operators and airline companies were able to redesign inclusive experiences for FwASD as a result of their reciprocal collaboration. Similar to a previous review [8], we found that valuable ideas from the stakeholders were tested during the airport visit and were shown to generate positive outcomes and, ultimately, a win–win relationship based on reciprocal learning and understanding with benefits for both the FwASDs and the airport operators and airline companies. For FwASDs, the benefits are the improvements in service delivery during critical encounters, while the airport operators and airline companies benefit from improving their social responsibility impact and corporate responsibility image.

Knowledge about how to manage the value co-creation process in the design of airport experiences can help airport operators and airline companies to more successfully identify adjustments and allocate the resources needed to design successful airport experiences for challenging markets.

## 5. Conclusions: Practical Implications

The results of this study can be used by airport operators and airline companies as a managerial tool to help redesign encounters [57,58] with FwASDs. In doing so, airport operators and airline company managers can react more quickly and ensure the timely, successful redesign of inclusive service delivery, which can also be applied in times of health crises such as the current COVID-19 pandemic. Our findings confirm that, when managing a co-creation process, it is crucial to identify the resources that need to be delivered [17] by the airport operators and airline companies in order for participants with communication and learning difficulties and their families to be able to derive the maximum value outcome from their participation in the design process [26]. Adapted communication tools must therefore be adapted before their implementation [36] to avoid negative outcomes [53].

The results of this study could be utilized in a number of ways. First, they can assist airport operators and airline company providers in developing new technological and non-technological products and services, such as the emerging virtual experiences to improve the air travel experience for FwASDs. Second, they can help airport operators and airline company managers allocate adequate resources to the design of successful airport experiences with and for FwASDs. Finally, they provide a novel opportunity to link service design to the satisfaction of FwASDs.

The implementation of new products and the services and facilities of airport operators and airline companies, along with adjustments to these, should be designed with the prior involvement of FwASDs and other stakeholders.

As with any study, the authors recognise the limitations of this research, as qualitative data was collected from one destination and limited to before and during the airport experience. These results must be carefully considered as a first approach to service-dominant logic co-creation processes with FwASDs in order to reduce barriers in airports for this group.

Given the heterogeneity of people with ASD and FwASD experiences, the study may have only captured a subset of issues impacting these families with regard to airport accessibility. It is also important to underscore that these types of changes, while focused on FwASDs in this paper, have larger, likely positive implications for the general population’s experience with air travel and the experience of those who have other conditions (e.g., anxiety and other neurodevelopmental disabilities).

## Figures and Tables

**Figure 1 ijerph-18-07206-f001:**
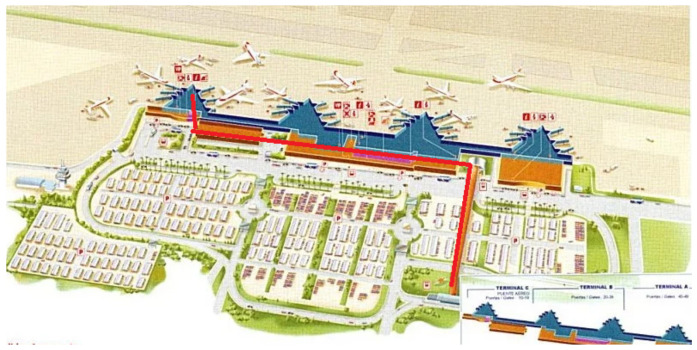
Airport visit experience from the parking area to the inside of the aeroplane. Terminal 1.

**Figure 2 ijerph-18-07206-f002:**
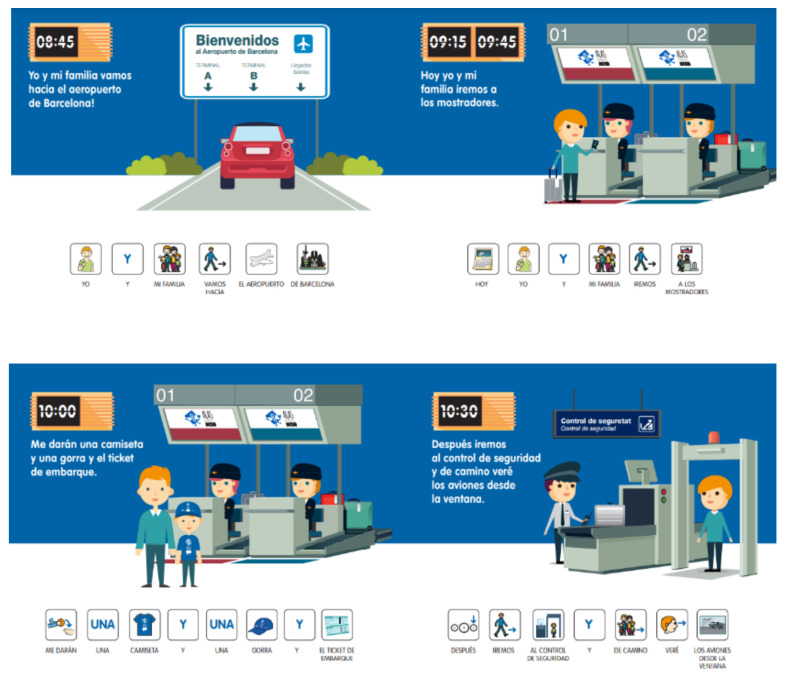
Storybook proposed by the focus groups and provided to FwASDs for preparation prior to the visit.

**Table 1 ijerph-18-07206-t001:** General characteristics of the sample of FwASDs (*n* = 25).

Variables	Categories	%
Gender of the child	Male	68%
	Female	32%
Frequency of use of air transport by the FwASDs	Habitually	0%
	Occasionally	7%
	Rarely	16%
	Never	77%
Usual mode of transport of the FwASD when travelling	Aircraft	22%
	Train	32%
	Car	46%
Age of child	From 3 to 9	60%
	From 10 to 15	35%
	Older than 15	5%
Need for air travel	Leisure	75%
	Visiting family	25%
Monthly household income	Less than EUR 1000	6%
	EUR 1000–2999	75%
	More than EUR 3000	19 %
Severity of disability	Less than 33%	12%
	34% to 65%	68%
	More than 65%	20%

FwASD—families of children with autism spectrum disorder.

**Table 2 ijerph-18-07206-t002:** Coding process.

Open Coding	Axial Coding	Main Themes (Selective Coding)
“The airport security staff were sensitive to the attitude of our 6-year-old child with ADS who was not willing to put his iPad through the security machine. This was comforting and put my child and the family at ease”. “We are not sure about drawing attention to our child’s disability since airport staff and airlines used to make us sign a liability form for security reasons, so we feel we don’t want to be excluded […]”.	Social interaction with airport staff and others’ attitudes.	Interaction with airport staff and other passengers (critical service encounter)
“The FwASD were given the storybook in advance of the visit, and it was very useful for anticipating the activity with my child in the days before the activity took place. The storybook showed what was going to happen during our airport visit […]”. “We would like to have timely, online information about delays and how to access the special service support because we find it difficult to get this information online”.	Adapted communication resources and use of new technologies	Importance of timely adapted information provision and resources for travel anticipation: online anticipation resources, storybooks, videos, booking information, and service support information (critical communication encounter).
“We provided new ideas during the focus groups and we were delighted when these were implemented in the airport visit, such as the use of quiet rooms and priority access to check-in and boarding”.	Universal design	Importance of universal design and proper facilities for accessibility, such as the availability of sensory rooms, priority check-in, etc. (critical usage encounter).

FwASD—families of children with autism spectrum disorder, ADS—autism spectrum disorder.

**Table 3 ijerph-18-07206-t003:** Inclusive resources.

Inclusive Resources	Critical Encounter Type	Value Outcome for FwASD
Storytelling in adapted format	Communication	Positive
One-day awareness training for 20 airport and aircraft staff members involved in the airport visit experience (3 April)	Service	Positive
Design of an easier wayfinding system to navigate through the airport	Service	Positive
Design of a quiet room available to FwASDs	Usage	Positive
Design of priority check-in and priority access through security control	Usage	Positive

FwASD—families of children with autism spectrum disorder.

## Data Availability

Data is partially contained within the article. While we included a number of photos in the article, they do not include pictures of the 50 respondents to avoid their identification, as they are a vulnerable group.

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
