# Peer review of "Autism Spectrum Disorder (ASD) and the Family Inclusive Airport Design Experience"

_ijerph, 2021, doi:10.3390/ijerph18137206_

Round 1
Reviewer 1 Report
This study utilized a S-D Logic approach in order to gain input into the accessibility of airports for FwASD. This topic is certainly one of interest to FwASD and is needed in order to make travel more accessible to neurodiverse populations. This paper provides helpful information, but requires revision.
Introduction
This section is generally well-written and laid out but I think could be improved by highlighting how the core symptoms of autism, in particular deficits in communication and the presence of sensory processing difficulties make navigating airports and travel particularly difficult. There will then be a more direct link to results section.
Materials/Methods
S-D Approach-This could be more extensively outlined in the paper so the reader knows exactly how that was applied here.
Figure 1-This does not seem to provide more illustration than what is written in the narrative. The bow below Phase 2 is also confusing and does not seem to be clarified by what is in the narrative. Specifically "1, 30 hour airport experience" is unclear as to what that means. Additionally, FwD is used in this illustration. It is unclear whether this was a typo meaning to be FwASD as used throughout the paper or if this is in reference to something new. That FwD abbreviation is also used in other places throughout the paper, so clarification on that would be helpful to the reader.
Table 1-There seems to be some misalignment. It would be helpful to the reader if the set up this table made the rows are easier to read and understand which of the the category/variable/percent go together.
Table 2-Similar to above. Hard to read clearly which Open Coding and Axial Coding go together.
Results
Table 3-This table does not help the reader visualize more than if this was written in narrative form in the Results section. If there is a way to improve this table so that it provides additional information useful the reader's understanding of the findings.
Line 335-There needs to be some revision of this sentence to provide a more explicit introductory statement about what you are setting up in that next section to review. Additionally, on line 362- an a) appears before Usage Critical Encounters but it looks like that list should have started above with Communication Critical Encounters on line 337.
Service Critical Encounters/Usage Critical Encounters Sections-It would be helpful to provide the reader with an example of the "wayfinding" that was developed for FwASD so that others reading this article may have a starting point if applying this work to their facility. The same can be said of the training provided to airport staff. It would be useful to outline what was covered in the training so that others who want to take this work and apply it might be best equipped. The social story that was included prior to these sections was extremely helpful. So anything like that to serve as examples in the sections following would help the reader to visualize these types of supports.
Discussion/Conclusions
The limitations of the study are missing and important to highlight, such as small sample size. Potentially, given heterogeneity of ASD and FwASD experiences the study may have only captured a subset of issues impacting these families with regard to airport accessibility. It is also important to underscore that these types of changes, while focused on FwASD in this paper, have larger likely positive implications for the general population's experience with air travel and those who also have other conditions (e.g., anxiety, other neurdevelopmental disabilities).
Author Response
Dear reviewer 1
Pleasesee the attachment

Reviewer 2 Report
Overall, this is a much needed study on a topic of much use to families of children with ASD and other disabilities. The writing needs to improve to increase ease of readability and here are a few suggestions towards that.
- The introduction needs to be a bit more crisp, concise and to the point. Consider using WHO recommendation on line 44 to be the opening statement (hook line), instead of starting that there is limited research on travel of people with psychiatric disorders.
- Consider rewriting the following sentence (Line 74) "This study aims to address this deficit to help improve the airport experience for FwASD." to something like "The aim of this study is to address this deficit by....
- “Project name hidden for anonymity” In an open review process, whats the purpose of this blinding?
- Consider reducing the acronyms. Its hard to keep track of them for example what is APT, S-D etc...
- Table 1, consider adding lines between categories to increase readability of numbers to subgroups. Now it is hard to decipher which group had what percentage.
- Results section seems to be too long when compared to the main findings in the discussion.
- I suggest summarizing main lessons learnt form the study to increase the value of study findings for airport staff and for families. Consider adding a different sub section "Practice implications" of the things currently mentioned in conclusion.
- Please address how threats to validity was addressed in this data collection & analysis process.
- Also, please acknowledge the limitations of this study.
Author Response
Dear reviewer 2
Please see attachment

Round 2
Reviewer 1 Report
Thank you for your efforts on these revisions! These provide additional clarity and underscore the importance of this work.